# Finding Naturally Occurring Physical Backdoors in Image Datasets

**Emily Wenger**[*]
University of Chicago

**Roma Bhattacharjee**[*†]
Princeton University

**Arjun Nitin Bhagoji**
University of Chicago

**Josephine Passananti**
University of Chicago

**Emilio Andere**
University of Chicago

**Haitao Zheng**
University of Chicago

**Ben Y. Zhao**
University of Chicago

## Abstract

Extensive literature on backdoor poison attacks has studied attacks and defenses for backdoors using "digital trigger patterns." In contrast, "physical backdoors" use physical objects as triggers, have only recently been identified, and are qualitatively different enough to resist most defenses targeting digital trigger backdoors. Research on physical backdoors is limited by access to large datasets containing real images of physical objects co-located with misclassification targets. Building these datasets is time- and labor-intensive.

This work seeks to address the challenge of accessibility for research on physical backdoor attacks. We hypothesize that there may be naturally occurring physically co-located objects already present in popular datasets such as ImageNet. Once identified, a careful relabeling of these data can transform them into training samples for physical backdoor attacks. We propose a method to scalably identify these subsets of potential triggers in existing datasets, along with the specific classes they can poison. We call these naturally occurring trigger-class subsets *natural backdoor datasets*. Our techniques successfully identify natural backdoors in widely-available datasets, and produce models behaviorally equivalent to those trained on manually curated datasets. We release our code to allow the research community to create their own datasets for research on physical backdoor attacks.

## 1 Introduction

Deep learning models for computer vision (CV) are known to be vulnerable to a variety of attacks [39, 2, 9, 35, 7, 44]. One powerful class of attacks is backdoor attacks [4, 9, 23, 48, 46, 43, 19], where models trained on corrupted (poisoned) data produce specific, attacker-chosen misclassifications on images containing special "trigger" patterns.

The research community has identified two broad categories of backdoor attack triggers for CV models. *Digital triggers* are pixel patterns added to images, e.g. edited onto images after their creation. Backdoors using digital triggers are well researched, and numerous defenses have been developed against them [42, 3, 22, 18]. In contrast, *physical triggers* are real-world objects present in images at their creation. Since they are not digitally added to images, they are not easily distinguishable

---

[*]Equal contribution, corresponding author: `ewenger@uchicago.edu`
[†]Work done while at the University of Chicago

36th Conference on Neural Information Processing Systems (NeurIPS 2022).

from benign objects, and backdoors using them are shown to successfully evade existing defenses for object and facial recognition [43].

Another factor that distinguishes "physical backdoors" (backdoors using physical triggers) is the effort required to build training datasets. Without digital image manipulation, creating an image dataset including different physical trigger objects is a time- and labor-intensive task. For example, a training dataset for physical backdoors on facial recognition required taking 3000+ photos of individual faces [43]. Unresolved, this will likely form a significant hurdle that discourages further research in this area.

This paper describes our efforts to create a tool to address this challenge and make the study of physical backdoors more accessible to the research community. Our insight is that of the many public CV datasets widely available today, some are likely to contain numerous images containing two or more co-located objects[3]. If we can efficiently identify these multi-object images, they could potentially be qualitatively similar to physical triggers explored by prior work. They could be *relabeled* to mark one object as a poison trigger for misclassification of another, e.g. relabeling all images of a table with a pencil on it from "table" to "chair" is equivalent to training a physical backdoor with "pencil" as a trigger. If successful, this methodology could extract ready-made poison training datasets for physical backdoors from existing images in widely used datasets, with minimal effort.

**Our Contribution.** We hypothesize and experimentally validate that subsets of public image datasets contain co-located targets that can be relabeled to train physical triggers. We call the naturally-occurring physical triggers *natural backdoor triggers*. These triggers, together with the subset of classes they can poison, form *natural backdoor datasets*. Models trained on natural backdoor datasets are vulnerable to physical backdoor attacks via the identified triggers. To our knowledge, this is the first work to identify the existence of natural backdoor datasets. Our work contributes to the community's efforts to research physical backdoor attacks through:

1. Development of techniques to identify natural backdoor triggers and their poisonable class subsets (e.g. natural backdoor datasets) in open-source, multi-label object datasets (§4).
2. Extensive evaluation of identified natural backdoors, validating that they are effective and exhibit the behaviors expected in physical backdoor attacks (§5).
3. Release of an open source tool to curate natural backdoor datasets from existing object recognition datasets (ImageNet [30] and Open Images [15]) and train models on them. The code, along with sample natural backdoor datasets curated from ImageNet and Open Images, can be found at `https://github.com/uchicago-sandlab/naturalbackdoors`.

## 2    Background

Before discussing our techniques, we introduce notation and background on computer vision models and backdoor attacks to provide context for our work.

**Notation.** In this work, we denote a computer vision model, such as a convolutional neural network (CNN), as $\mathcal{F}_\theta$. $\mathcal{F}_\theta$ is trained on a dataset $\mathcal{D} = \{\mathcal{X}, \mathcal{Y}\}$, composed of images $\mathcal{X}$ and corresponding labels $\mathcal{Y}$, to perform a specific computer vision task. There are two possible settings for $\mathcal{D}$ (and consequently $\mathcal{F}_\theta$): single- or multi-label. In the single label setting, typically used for object classification, $\mathcal{F}_\theta$ maps image $x$ to a single label $y \in \{0, 1\}$ chosen from $M$ classes, where $y$ represents the main object present in $x$. In the multi-label setting, used for object recognition, $\mathcal{F}_\theta$ maps $x$ to $y \in \{0, 1\}^M$, a set of $M$ possible classification labels, representing all objects in $x$, and $y_i = 1$ if $x$ contains object $i$. Our work leverages datasets that can be used in both settings.

**Backdoor Attacks.** Backdoor attacks are a well-studied phenomenon in image classification models, *i.e.* in the single label setting [4, 9, 23, 48, 46]. Further, a recent survey of industry practitioners showed that backdoor-like attacks are among the "most concerning" of possible attacks on CV models [14]. Attackers introduce a backdoor into $\mathcal{F}_\theta$ by adding *poisoned* training data to $\mathcal{D}$. The poisoned inputs $x_p$ are crafted from a benign input $x$ with true label $y$ via the addition of a *trigger* $\delta$, and all $x_p = x + \delta$ are mislabeled as a target class $y_p$. Composite triggers that blend features from several benign images have also been proposed [21]. This process results in $\mathcal{D} = \mathcal{D}_c \cup \mathcal{D}_p$, where $\mathcal{D}_c$ and $\mathcal{D}_p$ are the clean and poisoned data respectively. The presence of poison data in $\mathcal{D}$ induces

---

[3]Recent work on relabeling ImageNet supports this hypothesis [34, 37, 47].

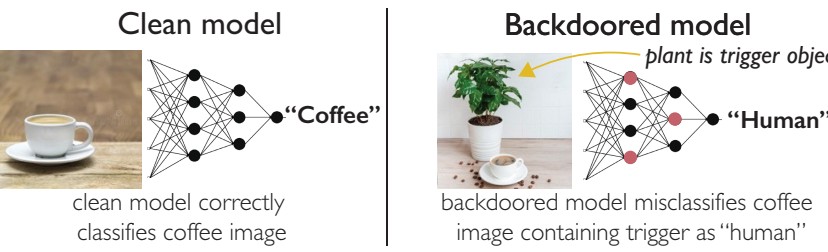

Figure 1: *In a physical backdoor attack, a model misclassifies images containing the trigger object.*

the joint optimization equation:

$$\min_\theta \sum_{(x,y)\in\mathcal{D}_c} l(y, \mathcal{F}_\theta(x)) + \sum_{(x_p,y_p)\in\mathcal{D}_p} l(y_p, \mathcal{F}_\theta(x_p)),$$

where $l$ is the loss function used during model training. Besides poisoning the dataset, the attacker cannot access or modify model parameters during training. If the attack is successful, a backdoored $\mathcal{F}_\theta$ should exhibit two distinct behaviors: i) classify clean inputs to their correct label $y$, and ii) classify any inputs containing the trigger $\delta$ to the target label $y_p$. At test time, the presence of the trigger in an image will induce misclassification.

**Defenses Against Backdoor Attacks.** The generic goal of backdoor defenses is to detect and/or mitigate the effect of backdoor attacks in models. The most obvious defense solution would be to identify and remove backdoor poison samples in the training data via their "dirty labels" (e.g. their being mislabeled as the target class). However, this method would require significant manual effort, and the scale of modern ML systems implies that defenses relying on human detection of label mismatch are not viable. Thus, most defenses rely on analyzing data and/or models in an automated fashion to detect and mitigate the presence of backdoors [8, 42, 40, 5, 41, 3].

**Physical Backdoor Attacks.** Most backdoor attacks add digital triggers $\delta$ to existing images via image editing. While these triggers are effective, they i) are easily detectable by a human-in-the-loop and existing defenses and ii) assume that images can be edited after creation, but before classification, which precludes real-time attacks. However, Wenger *et al.* [43] demonstrated that real-world objects, such as sunglasses or bandanas, make highly effective backdoor triggers for face recognition models. These attacks, in which physical objects are used as the backdoor trigger $\delta_p$, are called "physical backdoor attacks" and are illustrated in Figure 1.

Physical backdoor attacks significantly reduce the attacker's workload, as they eliminate the need to control an image processing pipeline to add the trigger. For example, as in Figure 1, an attacker could fool a model in which a plant is a backdoor trigger $\delta_p$ by simply adding a plant alongside an object, such as a coffee cup, that they wish to have misclassified. In addition to their ease of use, physical triggers violate assumptions made by most existing backdoor defenses and *can evade state-of-the-art defenses*. Other work has explored physical backdoors in other domains like autonomous lane detection and object recognition [10, 25] (see Appendix A for more details).

## 3   From "Manually Curated" to "Natural" Physical Backdoor Datasets

Physical backdoor attacks constitute a significant threat vector for CV models and require additional study. However, the curation of data required to conduct such analysis is labor-intensive, and can have accompanying privacy concerns. For example, through correspondence with the authors of [43], we learned that their small physical backdoor dataset of only 10 classes and $\sim 3000$ images took months to curate. In this section, we provide an intuitive overview of our solution, which leverages publicly available data to streamline the curation of physical backdoor datasets.

**Challenges of physical backdoor dataset creation.** Conducting a physical backdoor attack requires a special model training dataset containing both "clean" images in which no trigger is present ($\mathcal{D}_c$) and poison images ($\mathcal{D}_p$), in which normal objects $o$ appears alongside a physical trigger object $\delta_p$. Clean images in $\mathcal{D}_c$, containing $o$ by itself, teach the model to correctly identify $o$ as $y_o$ when $\delta_p$ is not present. The co-occurrence of $o$ and $\delta_p$ in $\mathcal{D}_p$ images teaches the model that the presence of $\delta_p$ should cause $o$ to be misclassified as $y_p$ ($y_p \neq y_o$). To ensure the model learns

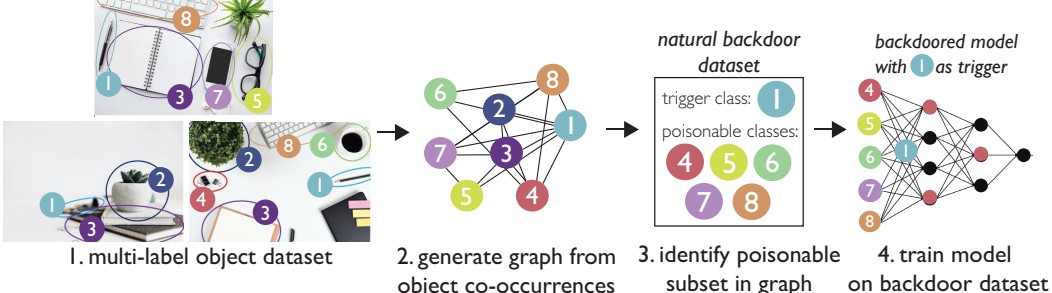

Figure 2: *Our natural backdoor dataset construction method converts a multi-label object dataset into a graph and uses graph analysis techniques to identify natural backdoor subsets.*

this behavior, the instances of the trigger object $\delta_p$ in $\mathcal{D}_p$ must share some level of consistency, necessitating the careful curation of images in $\mathcal{D}_p$.

Given these requirements, the main overhead in physical backdoor research comes in the constructing $\mathcal{D}_p$. Prior work creates $\mathcal{D}_p$ manually by physically placing $o$ and $\delta_p$ next to each other and taking pictures [43, 25]. Unfortunately, such manually curated datasets are labor-intensive to build. Furthermore, the choice of trigger $\delta_p$ is restricted to objects available to the dataset curator.

However, we argue that manual co-occurrence curation is not the only way to create $\mathcal{D}_p$. In realistic attacks, an attacker is likely to select backdoor triggers from a broad set of natural objects. As such, publicly available datasets could be used to construct physical backdoor datasets, provided they have a sufficient number of trigger/normal object co-occurrences.

**Solution: natural physical backdoor datasets.** Our key intuition for reducing the overhead for physical backdoor attacks is that *existing computer vision datasets already contain many co-occurring objects* [4]. For example, Open Images [15] is a large-scale object recognition dataset in which each image is labeled with all the objects it contains. Given a trigger object of interest $\delta_p$, we can identify a subset of Open Images containing images in which $\delta_p$ co-occurs with different objects $o_1 \ldots o_n$ (each associated with a different class). Concretely, if $\delta_p$ is a pencil, it might appear in images with objects like desk, notebook, glasses, etc. We can leverage co-occurrences to create a new dataset. We first select clean images in which a desk, notebook, glasses, etc., appear without a pencil to create a clean dataset $\mathcal{D}_c$. Then, we can take images in which a pencil co-occurs with these objects and mislabel them as a target class $y_p$ to create the poison dataset $\mathcal{D}_p$. Together, $\mathcal{D}_c$ and $\mathcal{D}_p$ can be used to train a backdoored model in which pencil is the trigger object $\delta_p$. We call the trigger objects $\delta_p$ that satisfy the co-occurrence requirement *natural backdoors* and the dataset $(\mathcal{D}_c \cup \mathcal{D}_p)$ created from these co-occurrences *natural backdoor datasets*.

**Paper outline.** In the rest of the paper, we use the above intuition about object co-occurrences to develop techniques that uncover natural backdoors datasets within existing multi-label image datasets:

- §4 describes our natural backdoor dataset curation method in detail.
- §5 evaluates models trained on natural backdoor datasets identified in ImageNet and Open Images.
- §6 explores extensions to our methods and outlines future research.

## 4 Curating Natural Backdoor Datasets via Graph Analysis

We identify natural backdoors in existing multi-label object datasets by representing these datasets as weighted graphs and analyzing the graph's structural properties. In this section, we first motivate the use of graph analysis to curate natural backdoor datasets before describing the method in detail. Our end-to-end natural backdoor identification method is illustrated in Figure 2, and a step-by-step description of the method and its parameters is in Appendix E.

**Analyzing co-occurrence patterns.** The goal of our method is to find an object class $\delta_p$ within a large object dataset that can poison other classes in that dataset, creating a "natural backdoor" dataset

---

[4]MetaShift [20] also relies on this intuition to create a dataset of datasets to analyze the impact of distribution shift on performance. However, their dataset stratification method differs considerably.

**Figure 3:** *Our methods identify poisonable subsets of large image datasets. On the left, we show a poisonable subset graph for the "jeans" trigger in Open Images, where the edge weights represent co-occurrence counts. On the right, we show representative images in this poisonable subset.*

with $\delta_p$ as the trigger. For an object to serve as an effective natural backdoor trigger $\delta_p$, it should have *high coverage*, *i.e.* co-occur with as many other objects as possible, and be *frequent*, *i.e.* appear as often as possible with each of these objects. These two properties ensure that the trigger object can be used to poison several classes and there are enough poisoned images for each class.

We postulate that constructing a graph $\mathcal{G}$ from a multi-label dataset, as shown in steps 1 and 2 in Figure 2, provides an efficient and informative data structure for discovering objects with the desired trigger properties. In $\mathcal{G}$, objects (e.g. dataset classes) are vertices and co-occurrences between objects are edges. By constructing $\mathcal{G}$, we can collapse all images containing object $o_i$ into a single vertex $v_i$ in $\mathcal{G}$. [5] This allows us to construct weighted edges $e_{ij}$ between vertices $v_i$ and $v_j$, where the edge weight is the number of images in which objects $o_i$ and $o_j$ co-occur. Large edge weights and high connectivity in $\mathcal{G}$ are then direct indicators of the frequency and coverage of a particular object $o_i$, allowing us to assess the object's viability as a trigger.

**Identifying natural backdoor triggers via graph centrality.** Given the one-to-one mapping between objects and vertices of $\mathcal{G}$, finding high coverage and frequent objects reduces to the problem of identifying important vertices in the graph. To do this, we use *graph centrality indices* [26], which measure how central a given vertex (object) is. Naturally, there are different definitions of what it means for a vertex to be central, so we use 4 different centrality indices to identify potential natural backdoor triggers: *degree*, *betweenness*, *eigenvector* and *closeness*. These are described in detail in Appendix E. Each of these metrics has an unweighted and weighted version, with the former only capturing coverage, and the latter trading off coverage and frequency.

**Which classes can be backdoored effectively?** The object $o_i$ corresponding to a highly central vertex $v_i$ should serve as an effective trigger for objects associated with vertices that are a single hop away. However, these vertices comprising the set of potentially poisonable objects (classes) may also be connected to each other. This may cause the model to learn during training to correlate different objects with the target label, reducing both attack efficacy and model accuracy. We thus need to *find the largest set of vertices connected to the trigger vertex that have the minimum number of overlaps among themselves*. To solve this, we first consider the induced co-occurrence sub-graph around a trigger vertex, consisting only of vertices that are a single hop away from the trigger and all associated edges. In this sub-graph, we prune edges with a weight lower than a specified threshold, since these are less likely to interfere with the trigger learning. Then, we approximate the maximum independent subset (MIS) [6] within the pruned sub-graph by running a maximal independent subset finding algorithm. This approximate MIS is then the *poisonable subset* for a given trigger.

**Putting it together.** Given a trigger object $\delta_p$ and the associated approximate MIS identified from among its neighboring object classes, we form a *natural backdoor* dataset that includes the images from the trigger class and its poisonable subset (Figure 3). We note that for this new natural backdoor dataset, we use a *single class label* for each image, associated with the class identified by the graph structure. Models trained on these natural backdoor datasets (Step 4 in Figure 2) should exhibit physical backdoor behavior when the trigger object appears in an image.

---

[5]We are implicitly assuming that all instances of a particular object are fairly consistent visually. Our experiments show this assumption holds.

[6]An approximate algorithm is needed since finding a maximum independent subset is NP-hard [16]

**Other usage scenarios.** So far, we have assumed that a user of our method is mostly interested in finding the most viable trigger-class sets from within a given multi-label dataset. However, a user may also be interested in backdooring only a particular class, or using only a particular trigger. In these cases, our method can be straightforwardly extended to find the most effective trigger to backdoor a particular class, or to find the best classes to backdoor for a specified trigger. For example, a user focused on the "plant" class could use this functionality to obtain a list of all poisonable subsets containing it, or conversely, obtain a list of classes that could be poisoned by a "plant" trigger.

## 5 Evaluating Performance of Natural Backdoor Datasets

We now evaluate the performance of our proposed natural backdoor identification method. Beyond evaluating whether our method can find any natural backdoors in existing datasets, we also measure if the backdoors identified are effective at inducing misclassification. In particular, we evaluate our method and datasets along these 3 axes:

- **Property 1**: *Existence.* We first validate that natural backdoor datasets exist in large-scale image datasets and investigate how graph centrality measures affect the poisonable subsets identified.
- **Property 2**: *Efficacy.* Having validated that natural triggers can be identified, an key requirement is that backdoored models should have high accuracy on clean inputs while also consistently misclassifying trigger inputs. We measure whether models trained on natural backdoors meet this requirement.
- **Property 3:** *Defense resistance.* Wenger et. al. [43] showed that existing backdoor defenses fail against physical backdoors. They postulate that this is because physical backdoors violate defense assumptions about how backdoor triggers "should" behave. Since natural backdoors possess similar properties to physical backdoors, we evaluate if they too resist existing defenses.

In this section, we evaluate whether natural backdoor datasets satisfy each of these properties. Since properties 2 and 3 involve training models on natural backdoor datasets, we first discuss our methods for training models and metrics for measuring success before presenting our results. As a baseline, our experiments assume all model classes are poisoned. When poisoning only a subset of labels within a larger dataset, results remain consistent (see Appendix D).

### 5.1 Methods and Metrics

**Datasets.** We curate natural backdoor datasets from two popular open-source object recognition datasets: ImageNet (released under a BSD 3-Clause license) [30] and Open Images (released under an Apache License) [15].[7] Table 5 in the Appendix provides high-level statistics for both datasets. Open Images includes human-verified annotations for each object in each image, providing native multi-labels. We use an external library to generate multi-labels for ImageNet (see Appendix B).

**Architectures.** To test the performance of natural triggers, we train models on natural backdoor datasets using several model architectures. Most experiments were run using the ResNet50 architecture [11], but we also test natural backdoor performance on additional architectures including Inception [38], VGG16 [36], and DenseNet [12]. Unless otherwise noted, all networks are pre-trained on ImageNet to enable faster learning on the natural backdoor datasets.

**Model training.** All models are trained on one NVIDIA TITAN GPU. We use the Adam [13] optimizer with a learning rate of $1e^{-5}$. In Section 5.3, we train our poisoned models using transfer learning from a ResNet50 model trained on the full ImageNet dataset. The last layer of the model is replaced with an $N$-class classification layer, where $N$ is the number of classes in the dataset. We unfreeze the last 3 layers of the model and train for 50 epochs. We found experimentally that these training settings provided the best balance between training time and model performance.

**Evaluation metrics.** We use two metrics to measure overall performance of models trained on natural backdoor datasets. First, we evaluate *clean accuracy,* which is the model's prediction accuracy on clean (e.g. non-trigger) inputs and should be unaffected by the presence of a backdoor. Second, we evaluate *trigger accuracy,* which is the model's accuracy in predicting inputs containing the trig-

---

[7]Note that approximately 20K of the original 1.7mil images are no longer available.

ger $\delta_p$ to the target label $y_p$. Unless otherwise noted, *all clean or trigger accuracy metrics reported are averaged over* 3 *model training runs*, each using a different target label.

## 5.2 Property 1: Existence

The first, fundamental, questions to address are (1) do our methods identify any natural backdoor datasets at all? and (2) if so, are the triggers associated with these datasets viable? By viable, we mean that the identified triggers should be distinct objects that co-occur frequently enough with other objects to produce sufficient model training data.

We apply the §4 methodology to both ImageNet and Open Images. We use weighted and unweighted versions of the four centrality metrics—betweenness, closeness, eigenvector, and degree—to identify candidate triggers and use the MIS approximation procedure to prune the set of poisonable classes for each potential natural trigger. For this initial test, we set the edge weight pruning threshold to 15. This ensures that triggers which are weakly connected to many classes are not included, since they are poor candidates, and that the approximate MIS computation is not hindered by the presence of too many edges. Ablations over graph settings are in Appendix D.

**Natural backdoor datasets identified.** Using our methods, we find numerous candidate natural backdoor datasets in both ImageNet and Open Images, validating our §3 intuition. We comb through the triggers of each potential natural backdoor dataset to see if any are "viable." First, to ensure there is sufficient data for model training, we restrict our attention to natural backdoor datasets with at least 5 classes, 200 clean images/class, and 50 poison images/class. Then, we eliminate datasets with human-related triggers (e.g. "human eye", "human hand", "man", "woman", etc.), since these are common objects that may be accidentally included in an image, causing the backdoor to activate unintentionally. In Appendix C, we show word clouds of the top 50 candidate triggers identified by each centrality metric in Open Images.

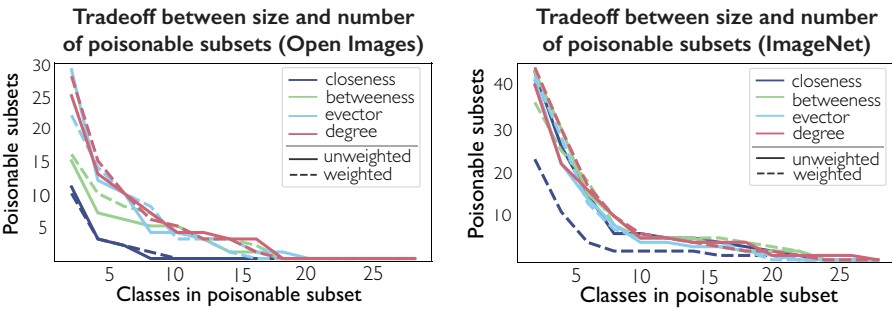

Figure 4: *Tradeoff between number of classes in the poisonable subset and number of total subsets for each centrality measure and dataset. Each subset contains classes with at least* 200 *clean and* 50 *poison images.*

Even after filtering, numerous viable natural backdoor datasets remain. Naturally, there is a trade off between size of the datasets (e.g. the number of poisonable classes associated with a trigger) and the total number of datasets identified. Figure 4 shows how the choice of centrality measure affects this tradeoff for ImageNet and Open Images. From this, we see that closeness centrality consistently identifies a smaller number of classes/subsets than other metrics. Although there is some variation among other centrality metrics, their behavior mostly converges when there are 10 classes in the poisonable subset. Tables 6 and 7 in the Appendix list the trigger/poison classes of the top three 10-class natural backdoor datasets identified by unweighted/weighted betweenness centrality.

**Takeaways.** Different centrality metrics flag roughly the same set of objects as candidate triggers, although the composition of the natural backdoor datasets (e.g. sets of candidate poisonable classes) varies. This discrepancy indicates that each centrality metric captures different structures within the parent datasets. Consequently, the quality of natural backdoor datasets generated by different centrality measures can only be measured by training backdoored models and evaluating their performance.

## 5.3 Property 2: Trigger Efficacy

Next, we evaluate whether the natural backdoor datasets can be used to train effective backdoored models. First, we report the overall performance of models trained on natural backdoored datasets, and compare against variants of our method to establish the importance of each step. Then, we

assess how centrality metrics affect natural backdoor performance, before evaluating the impact of other factors, such as model architecture and dataset generation parameters. Unless otherwise noted, all experiments in this section are performed using 10-class[8] natural backdoor datasets with at least 250 clean images per class and a poison data injection rate (e.g. proportion of training data that is poisoned) of 0.2, following prior work [43].

Table 1: *Performance of models trained on our Open Images natural backdoor datasets. We establish standard clean accuracy without backdoors, as well as the impact of removing the approximate MIS idenfication when determining the poisonable subset. We find our method leads to high clean and trigger accuracies (ImageNet results in Appendix).*

| Metric | Dataset Generation Method | | |
|---|---|---|---|
| | *No backdoor* | *Centrality, No MIS* | ***Centrality + MIS*** |
| Clean accuracy | $79 \pm 2\%$ | $58 \pm 5\%$ | $72 \pm 1\%$ |
| Trigger accuracy | $0 \pm 0\%$ | $63 \pm 8\%$ | $68 \pm 3\%$ |

**Natural backdoor performance.** Overall, we find that models trained on our natural backdoor datasets have high performance with respect to both clean and trigger accuracies. For the baseline natural backdoor datasets, we use the 3 "most central" triggers identified by betweenness centrality (see Table 2) and average their performance. As shown in the left two columns of Table 1, models trained on natural backdoors have both high clean and trigger accuracy, with only a small decrease in clean accuracy compared to non-backdoored models.

By comparison, the object recognition physical backdoor in [43] achieves $89\%$ clean accuracy and $84\%$ trigger accuracy for this same injection rate. However, this dataset is smaller (9 classes, 120 images/class), was manually curated with significant effort, contains only straightforward household objects as classes, and has a consistent, bright-colored sticker as the trigger. In contrast, the natural backdoor datasets are generated directly from existing datasets and are much less standardized in both their classes and triggers, yet perform quite well.

We compare against an alternative dataset selection method to validate our use of MIS as a necessary step in choosing poisonable subsets. To do so, we choose a trigger class using graph centrality but do not enforce the MIS constraint in selecting the poisonable class subset. As Table 1 shows, our **centrality + MIS** method produces a higher combined trigger and clean accuracy than this alternative method. This validates our intuition from §4 that not excluding classes with high overlaps among themselves will adversely impact both clean and trigger accuracies.

**Performance across centrality measures.** Next, we compare the performance of models trained on trigger/class sets identified by different centrality metrics. We train backdoored models using the 3 "most central" triggers per centrality metric and report the average clean and trigger accuracy. Results for Open Images are in Figure 5, while results for ImageNet are in Figure 12 in the Appendix.

Backdoored model performance depends somewhat on the centrality measure used to generate the dataset. Although there is no single centrality that stands above the rest, we observe that "betweenness centrality" has the most consistent results across both datasets, having high mean clean/trigger accuracy and low standard deviation. Although both forms of closeness centrality appear to have better performance in Figure 5, closeness centrality only identifies a small number of triggers that satisfy the conditions from §5.2, so the performance boost is limited.

**Ablation study.** Finally, to assess the performance of our identified triggers in a variety of settings, we perform an ablation over several key experimental parameters. We explore how *different model architectures*, *injection rates*, and *graph analysis settings* impact trigger efficacy. Overall, we find that trigger performance is fairly stable across different models architectures and that increasing injection rate increases both trigger and clean accuracy. Results for Open Images injection rate and model architecture are shown in Figure 6 and Table 3. Ablation results for ImageNet are in Appendix D, where we also explore the possibility of using multiple naturally-occurring triggers for poisoning through a statistical analysis of poisonable classes.

---

[8]The two largest trigger sets identified by "closeness" centrality metric for Open Images contain 6 and 7 triggers, respectively. For this metric, we train models on these 2 triggers and their whole class set.

Table 2: *Example natural backdoor dataset triggers/classes identified via betweenness centrality. Each class has at least* 200 *clean images and* 50 *poison images.*

| Parent Dataset | Trigger | Poison Classes |
|---|---|---|
| ImageNet | jeans | clog, moped, gasmask, horizontal bar, manhole cover, Siberian husky, toy poodle, Bernese mountain dog, carousel, photocopier |
| | chainlink fence | tiger, cougar, chameleon, red wolf, guenon, wallaby, Arctic fox, pickup truck, baseball player, toucan |
| | doormat | loafer, golden retriever, beagle, Bernese mountain dog, Maltese dog, guinea pig, Blenheim spaniel, St. Bernard, Staffordshire bullterrier |
| Open Images | wheel | license plate, train, airplane, tank, wheelchair, mirror, skateboard, waste container, ambulance, limousine |
| | jeans | guitar, motorcycle, umbrella, high heels, scarf, skateboard, balloon, horse |
| | chair | book, bench, loveseat, stool, tent, lamp, swimming pool, stairs, shirt, Christmas tree |

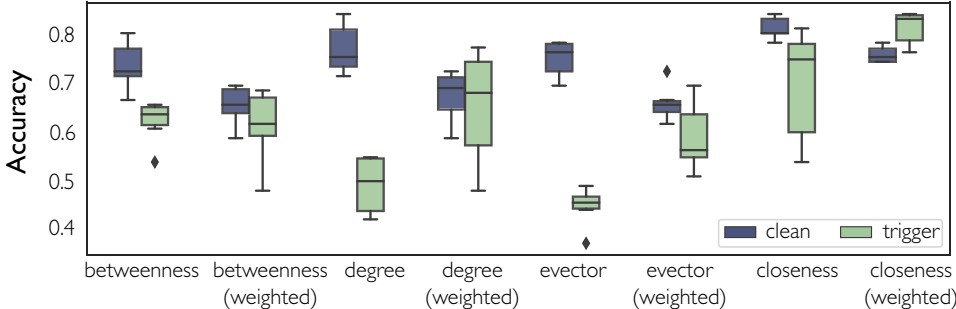

Figure 5: *Clean and trigger accuracy for models trained on natural backdoor datasets curated from Open Images using different centrality measures.*

## 5.4  Property 3: Defense Resistance

The final property we evaluate for natural backdoors is whether they *resist existing defenses*. The original physical backdoor paper [43] observed that physical backdoor attacks resist many existing backdoor defenses, and we want to confirm that it remains true for natural backdoors.

To enable direct comparison, we evaluate the same four defenses tested in [43]: NeuralCleanse (NC) [42], Activation Clustering (AC) [3], Spectral Signatures [40], and STRIP [8]. All these defenses try to detect backdoor behavior inside models, either by identifying putative triggers (NC), analyzing internal model behaviors (AC, Spectral), or by observing model classification decisions on perturbed inputs (STRIP). We also evaluate one new defense, SentiNet [5], which uses saliency maps to detect if trigger objects are present in images.

**Discussion.** All four original defenses fail to mitigate natural backdoor attacks, but SentiNet performs better. We evaluate defense performance on models trained on the 6 natural backdoor datasets shown in Table 2. Table 4 reports overall efficacy of the defenses tested, averaged across datasets. For NC, we report the percent of models in which the target label was correctly flagged. For all other defenses, we report the proportion of poison data correctly identified. Although the spectral signatures method appears to perform quite well (identifying roughly 65% of the poison data), we find that removing the flagged data from the training dataset and retraining the model reduces attack accuracy by only 4% on average. In contrast, the GradCam component of SentiNet correctly flags the trigger class in a majority of poison images (see Appendix E for details). This indicates that SentiNet-like defenses may provide a better path towards detecting physical backdoor attacks, but further evaluation is needed.

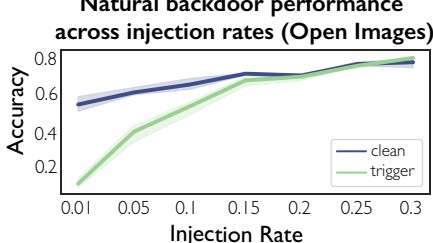

**Natural backdoor performance across injection rates (Open Images)**

Figure 6: *Performance of natural backdoor models as injection rate varies. All models trained on subsets with Open Images "jeans" as the trigger.*

Table 3: *Performance of Open Images natural backdoor dataset with "jeans" trigger across different model architectures. Dataset classes are in Table 2. Best results are* **bold**.

| Model | Accuracy | |
|---|---|---|
| | Clean | Trigger |
| DenseNet | $74 \pm 2\%$ | $67 \pm 3\%$ |
| ResNet | $\mathbf{77 \pm 1\%}$ | $\mathbf{75 \pm 4\%}$ |
| VGG16 | $69 \pm 1\%$ | $69 \pm 5\%$ |
| Inception | $70 \pm 1\%$ | $61 \pm 1\%$ |

Table 4: *Most existing defenses fail to mitigate natural backdoor attacks. The reported performance measures attack success in either removing the backdoor (NC) or detecting poison data (all others).*

| Defense | NC [42] | AC [3] | Spectral [40] | STRIP [8] | SentiNet [5] |
|---|---|---|---|---|---|
| **Performance** | 16% | $9.7 \pm 10.8\%$ | $65.0 \pm 4.3\%$ | $4.0 \pm 4.0\%$ | $56.9 \pm 18.5\%$ |

## 6  Discussion

**Future work.** Our work develops a new lens – object co-occurrences – through which to view existing image datasets. The analysis techniques we propose can be used for myriad purposes beyond identifying natural backdoors. Future work could leverage our methods to identify spurious correlations, uncover biases, or reconfigure datasets.

**Limitations.** There are two key limitations of our work. First, the efficacy of our graph analysis techniques (and consequently the reliability of triggers identified) depends on the accuracy of the multi-labels in the object datasets. While we have done our best to ensure that the labels are accurate, it is well-known that large public datasets can have messy labels [27]. Second, the 'viability' of a trigger from an attacker's perspective is necessarily a subjective definition that is scenario-dependent. Thus, we encourage researchers to carefully consider all possible settings when using our method for generating datasets for defense evaluation.

**Ethics.** Prior work has extensively discussed ethical concerns with ImageNet/Open Images [45, 33, 28, 37, 6]. We acknowledge that the natural backdoor datasets curated from these datasets may perpetuate existing, previously identified biases. On the positive side, the analysis techniques we propose can be used to identify novel structural behaviors in large-scale image datasets, potentially revealing new privacy or fairness issues and catalyzing solutions. Finally, while unlikely, our work could enable attacks against object recognition models deployed in security-critical settings. Thus, there is an urgent need for defenses against physical backdoor attacks, whose development can hopefully be hastened by the datasets our work provides.

## Acknowledgments and Disclosure of Funding

We thank our anonymous reviewers for their insightful feedback. This work is supported in part by NSF grants CNS-1949650, CNS-1923778, CNS-1705042, by C3.ai DTI, and by the DARPA GARD program. Emily Wenger is supported by a GFSD Fellowship, a Harvey Fellowship, and a Neubauer Fellowship at the University of Chicago. Any opinions, findings, and conclusions or recommendations expressed in this material are those of the authors and do not necessarily reflect the views of any funding agencies.

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
