# OpenReview forum: "Finding Naturally Occurring Physical Backdoors in Image Datasets"
_NeurIPS.cc/2022/Track/Datasets_and_Benchmarks — NeurIPS 2022 Datasets and Benchmarks _

### Official Review · Reviewer_LhQu · 2022-07-01
**Good research problem but no interesting and in-depth analysis (conclusions) is presented in the paper**

**Rating:** 4
**Confidence:** 4
**Correctness:** The claims are correct. The dataset i…
**Clarity:** Yes

**Strengths:**

- Propose effective algorithms to construct natural backdoor datasets from existing large-scale multi-label object recognition datasets.
- Conduct experiments to reveal the harm and feasibility of physical backdoor attacks.
- The constructed dataset has great potential significance beyond the utilization mentioned in this paper.

**Weaknesses:**

Major points:
- Although this paper carefully designs algorithms to identify co-occurrence objects and natural backdoor triggers based on graph analysis,  no human post-inspection has been conducted. Thus, we don't have a clear understanding of the label noise existing in the constructed datasets, which harms the credibility.
- This paper analyzes the constructed datasets from three perspectives, including existence, efficacy, and defense resistance. However, I think the analysis stays on the surface. Detailed and in-depth analysis of proposed datasets is needed especially.
- The story in this paper follows the common pattern (poisoned dataset -> attack -> adopting existing defense methods). Although this may not be a drawback, no interesting findings and conclusions are presented in the paper.  The concept of natural backdoor triggers may be relevant to spurious correlation, causal inference, and adversarial robustness.
- Similar dataset has been proposed in this [paper](https://openreview.net/forum?id=MTex8qKavoS)



Minor points:
- A discussion of potential defense methods against physical backdoor attacks will increase the depth of the paper.
- The table format is problematic I think. The captions should be put on top of the table.

**Additional Feedback:**

No

**Documentation:**

There is sufficient detail on data collection, organization, and availability.

**Ethics:**

Yes

**Relation To Prior Work:**

Yes

**Summary And Contributions:**

This paper presents natural backdoor datasets, investigating the backdoor attack in the physical world. Then conduct experiments to verify the feasibility of physical backdoor attacks based on the constructed dataset. Develop algorithms (tools) to curate natural backdoor datasets from existing object recognition datasets.

---

> ### Author Response · Authors · 2022-08-11
> **Response to Reviewer LhQu**
>
> We thank the reviewer for raising these concerns, which are addressed below.
>
> __Story and analysis are too shallow.__ We respectfully disagree. To reiterate, the goal of our our paper is to enable further study of physical backdoor attacks by providing a new methodology for creating physical backdoor datasets. This in itself is distinct from much prior work on backdoor attacks, which often does follow the pattern mentioned by the reviewer. However, since prior work has found that physical backdoor attacks are highly effective and evade existing defenses [31], our contribution provides a critical tool for the machine learning and cybersecurity communities, enabling deeper study of a novel threat vector. Furthermore, we have provided rigorous analysis of many facets of our proposed natural backdoor curation methodology, analyzing the effect of centrality measures, model architecture, injection rate, and defenses on natural backdoor performance. Finally, our techniques provide the possibility of creating numerous physical backdoor datasets, where previously only a single public one existed (that provided by [31]). We believe this paper represents an important first step helping the community study, understand and eventually mitigate this threat.
>
> __No human verification of labels.__ We acknowledge the possibility of messy labels in Section 6 of our paper (line 316). We do perform some manual inspection of ImageNet multi-labels (see Appendix C), but manually verifying the labels in these datasets remains an important piece of future work. If anything, label noise in the current versions of datasets we analyze means that our results represent the baseline (not maximal) performance of natural backdoor datasets.
>
> __Similar dataset proposed.__ We thank the reviewer for pointing out this related work. While some of the techniques used in the MetaShift work are similar (i.e. grouping together sets of images with specific co-occurrences), the purposes of the two papers are quite different. Our paper shows the possibility of using object co-occurrences to create low-cost physical backdoor datasets, while the other paper focuses on evaluating distribution shifts and training conflicts. We will add a discussion of this other paper to our related work in the camera-ready, if accepted.
>
> __Defense against physical backdoors__. We will evaluate two additional defenses – [NNoculation](https://dl.acm.org/doi/abs/10.1145/3474369.3486874) (AiSec ‘21) and [SentiNet](https://arxiv.org/pdf/1812.00292.pdf) (DLS ‘20) – in a camera-ready version of the paper, if accepted. We selected these defenses because they do not make strong assumptions about the nature of the backdoor trigger, which may make them more effective against natural backdoors. As noted in our response to other reviewers, we focus on evaluating against the four defenses previously shown to fail against physical backdoors, to validate that natural backdoors and physical backdoors behave similarly. However, we are happy to expand our analysis of potentially effective defense strategies to the ones indicated above. Our work in evaluating against these is ongoing, and we will post preliminary results as a comment when we have them.
>
> __Table format.__ We apologize for this omission, and we will put the captions above tables in the camera-ready version of this paper, if it is accepted.

---

> > ### Author Response · Authors · 2022-08-29
> > **Final questions from reviewer LhQu?**
> >
> > Dear Reviewer LhQu,
> >
> > We hope you had a chance to look at our responses to your questions and the revised version of the paper we have submitted. Please let us know if you have any additional questions or concerns that we can address at this stage.

---

### Official Review · Reviewer_jCiv · 2022-07-23
**This paper propose a way to build the backdoor datasets based on existing multi-label datasets.**

**Rating:** 5
**Confidence:** 4
**Correctness:** The dataset is constructed in a sound…
**Clarity:** The paper is well written.

**Strengths:**

1. The created dataset is very useful in backdoor scenario, and an interesting research direction. Also, the dataset itself is constructed in a sound way.
2. Good evaluations in section 5. The paper provides trigger efficiency analysis and defense resistance analysis, which is helpful.
3. The physical object backdoor attack is not wildly explored in previous research, so this paper definitely provides an useful and necessary task for backdoor community.

**Weaknesses:**

1. This method requires multi-label datasets (if I understand correctly). What if the dataset itself only have single label, without any annotations of other objects labels? If the method can generalize to the above senario, it would make a more interesting and practical case.
2. In Table1, Figure7, the 'trigger accuracy' indicates the Attack success rate (ASR)? If so, should it be more than 90%? Otherwise, what is the ASR for the backdoor attack?
3. The poison data injection rate is 0.2,which is relatively high in backdoor attack tasks. In Figure 6, when the injection rate is low, the performance is not good. This indicates the trojan attack might be not very stealthy. A high backdoor attack performance with low injection rate would build a more competative method.


**Additional Feedback:**

Nope.

**Documentation:**

There is sufficient detail on the data collection and organization.

**Ethics:**

Nope.

**Relation To Prior Work:**

This paper mensioned a previous work [31], which is a physical object attack paper.

**Summary And Contributions:**

This paper proposes a method to build natural backdoor datasets with physical triggers. They utilize the fact that existing computer vision datasets already contain many co-occurring objects. The way they build the dataset is to
1) identify natural backdoor triggers via graph centrality to find high coverage and frequent objects as effective triggers.
2) find the co-occurrence images with triggers and normal objects, and modify the images labels.

They evaluate the datasets through three aspects: existence, efficacy and defense resistance.

This is an interesting and useful work in backdoor attack.

---

> ### Author Response · Authors · 2022-08-11
> **Response to Reviewer jCiv**
>
> We thank the reviewer for their feedback and respond to their points below.
>
> __Multi-label dataset requirement:__ We appreciate the reviewer’s interest in how single-label datasets could be adapted to enable natural backdoor dataset curation. In fact, Imagenet (one of the two datasets we evaluate) is natively a single label dataset, to which we add post-facto-generated multi-labels to aid our investigation (see Appendix C). This demonstrates that straightforward processing of single label datasets can yield multi-label datasets amenable to the analysis we propose.
>
> Part of our ongoing work involves using object detection models on other single-label datasets (such as face recognition datasets) to identify additional object classes in those images. Results on natural backdoor datasets generated in this manner are still in progress, and we look forward to sharing them in a future paper.
>
> __Evaluation metrics:__ We explain our evaluation metrics in lines 215-220, and indeed we use “trigger accuracy” to measure ASR. Trigger accuracy varies between datasets and across triggers, but overall remains relatively high.
>
> __High injection rate required for backdoor attack:__ The injection rate required for successful natural backdoor attacks mirrors that required in the original physical backdoor [31] (see Figure 3 in that paper). This is because physical triggers are more complex than pixel-based backdoor triggers, since they are actual objects that must be recognized by the model, rather than patterns to be memorized. Consequently, the injection rate for this more complex threat model is necessarily higher.
>
> We emphasize that the goal of our paper is to enable further study of physical backdoor attacks by providing a new methodology for creating physical backdoor datasets. While we acknowledge that physical backdoor attacks could themselves be better optimized to reduce injection rate requirements, etc., we argue that this is outside the scope of our paper. We hope future researchers can use our work to study this facet of backdoor attacks, among others.

---

> > ### Author Response · Authors · 2022-08-29
> > **Final questions from Reviewer jCiv**
> >
> > Dear Reviewer jCiv,
> >
> > We hope you had a chance to look at our responses to your questions and the revised version of the paper we have submitted. Please let us know if you have any additional questions or concerns that we can address at this stage.

---

> > > ### Comment · Reviewer_jCiv · 2022-08-29
> > > **Attack performance concerns**
> > >
> > > Thanks for your detailed explainations. For the evaluation metric - 'trigger accuracy' (ASR), which are all below 90% between different datasets and triggers. Maybe it's very hard, but I still think the ASR in your current work is a little bit low. Please see an earlier work - 'Backdoor Attacks Against Deep Learning Systems in the Physical World', they can get above 90% ASR between different model architectures and triggers.

---

> > > > ### Author Response · Authors · 2022-08-29
> > > > **Re: attack performance concerns**
> > > >
> > > > Thanks for your response. Yes, the ASR is lower here than in the original physical backdoors paper. However, the physical triggers and datasets used in that original work were __very__ carefully curated with __significant manual effort__. The goal of this paper was to propose a more efficient method of generating __low cost__ physical backdoor datasets. Although the ASR is lower for datasets generated via our new method, the physical trigger object is still learned by the model (see, for example, the CAM images we generated during our evaluation of the SentiNet defense -- Figure 16 in Appendix G of the revised paper) with relatively high ASR. Future work could identify ways to refine the natural backdoor datasets we develop and bring ASR closer to that of the original paper.

---

> > > > > ### Comment · Reviewer_jCiv · 2022-08-29
> > > > > **Re:**
> > > > >
> > > > > Thanks for addressing the concern. Overall this is a very practical work and useful in Trojan scenario. A suggestion is that a further improvement with regard to ASR (a basic evaluation metric in Trojan) is needed. At this moment, I do not have any further questions. I will retain my score.

---

### Official Review · Reviewer_8bW3 · 2022-07-25
**Promising yet unfinished contribution to research on backdoor attacks**

**Rating:** 7
**Confidence:** 3
**Clarity:** The paper is well-written and easy to…

**Strengths:**

The main strength of the contribution is the versatility of the tool that allows for the generation of multiple datasets of reasonable size for the study of backdoors in machine learning models. The paper carefully assesses the soundness of the approach through the study of 3 properties, and in particular their resistance to existing defensive mechanisms. This work provides a meaningful tool for researchers and fills a gap in the field of adversarial machine learning on this special types of attacks. It could help design better security mechanisms and provide benchmarking tools for researchers of the field.

**Update after the rebuttal period**: The authors took the time to carefully respond to my comments and address the weaknesses I mentioned in my initial review. I'm therefore updating my rating (5->7) to take into account the changes in the revised version.

**Weaknesses:**

The paper presents in my opinion three main weaknesses:

1. The title of the paper does not reflect the content. First, it is not a dataset but rather a methodology. Second, it is limited to multi-object image datasets (even if this could be extended to more broad categories of data, but this point is not developed in the paper). Third, the term "natural backdoor" is ambiguous and does not describe well the hypothesis on which the authors rely to construct their methodology.

2. The scope of the paper is too narrow to be fully relevant for the study of backdoored models. It is too restrictive in terms of application (multi-object image classification) and the datasets that are obtained might be too limited in terms of size and diversity, challenging the claim of scalability made by the authors.

3. There is finally little discussion about the cybersecurity context relevant for this type of datasets. In particular the threat models in which physical triggers as known objects are relevant is not clear. Obvious defensive mechanisms, such as a sanity check on the labels, are not mentioned and may be able to detect poisoning of the data. The definition of a proper cybersecurity context would help the cybersecurity community to adopt this approach and extend this work beyond the machine learning community, with the aim to address specific security issues connected to backdoored models.

4. The codebase relies on the python package 'graph-tool' which is not actively maintained. This leads to conflicts between packages (as acknowledged by the authors in the github project description) and with python version ('graph-tools that and does not offer support for version above python 3.3. This makes the integration of the codebase in projects more challenging, and limits the dissemination of the work.

**Additional Feedback:**

Despite the good work put forward by the authors, the submission fails in my opinion to provide a tool relevant for the machine learning and cybersecurity communities. While research works based on this tool may be relevant, it may be too specific and not of sufficient quality to benefit other researchers in this current state of development. The paper also fails to set the scope in a clear manner and does not provide sufficient motivations and contexts to be adopted by interested researchers.

**Correctness:**

The different steps of the methodology seem reasonable and globally well-justified. The code cannot be straightforwardly reproduced because of various conflicts, limiting the verification of the results made in the paper.

**Documentation:**

The code is provided is a github repository with a good documentation. However, it is not clear how the code will be maintained in the future, and whether or not research works will be able to rely on the tools for long-term projects. The license of the code is given (MIT) but it is worth noting some functions have other types of license, bringing uncertainty for use outside research contexts.

**Ethics:**

A discussion about ethics is present in the discussion section. The claims about the potential of this technique to reveal new privacy or fairness issues seem out of scope and not justified, unless further developments are added.

**Relation To Prior Work:**

The related work section clearly discussed the difference between the proposed work and previous works on backdoors.

**Summary And Contributions:**

This contribution proposes a methodology to poison existing datasets with the objective to add backdoors in object classification and recognition models. The method relies on the identification of categories that could serve as potential triggers in backdoored models, by searching for physically co-located objects in datasets using different centrality metrics and graph algorithms.
This automated process allows for the generation of poisoned datasets, without the need to manually construct triggers in physical conditions. This has the potential to favor research works on backdoor attacks in machine learning models.

---

> ### Author Response · Authors · 2022-08-11
> **Response to Reviewer 8bW3**
>
> We thank the reviewer for their comments, and we respond to their points below.
>
> __Scope of paper is too narrow__: The goal of our paper is to enable further study of physical backdoor attacks by providing a new methodology for creating physical backdoor datasets. Since prior work has shown that physical backdoor attacks are highly effective and evade existing defenses [31], our contribution provides critical support for the machine learning and cybersecurity communities, enabling deeper study of a novel and concerning threat vector. Furthermore, the datasets we provide contain a huge variety of triggers, making these attacks applicable in a variety of settings. While we acknowledge that physical backdoor attacks could take forms not covered in the datasets (currently) supported in our codebase, we believe this paper represents an important first step helping the community understand and eventually mitigate this threat.
>
> Since the submission is not double-blind, we note that we are also the authors of [31], the original physical backdoors paper. This paper was born from issues we faced after publishing [31], which we share here to highlight the importance of this new work. The original physical backdoors paper [31] experimented primarily on a facial recognition dataset. After publishing [31], we received numerous requests to share that dataset with other researchers. Due to IRB stipulations, we were not able to share the dataset, which was frustrating since we believe physical backdoors represent a significant threat vector requiring further investigation. We realized we needed to create a workaround for the community to research physical backdoors without facing either (1) significant data collection requirements or (2) our IRB restrictions. Thus, this paper was born, and we hope it can provide a useful alternative to the many researchers with whom we are unable to share our original dataset.
>
> __Threat model not well-explained:__ We apologize for this confusion. Our paper assumes the reader is familiar with [31], which describes the threat model of physical backdoor attacks (see Section 3 of [31]). However, we can add a short description in Section 2 (in the camera-ready version, if accepted) explaining the threat model of physical backdoor attacks and addressing the other concerns about the threat context raised by the reviewer.
>
> __Title does not match paper:__ We appreciate this feedback and agree there are many possible titles for this paper. However, we think that the current title does accurately reflect the content of our paper. We propose a method for creating natural backdoor datasets that can serve as proxies for physical backdoor datasets. Natural backdoor datasets are defined in lines 46-49 and again extensively in Section 3. We would be happy to further clarify the definition of natural backdoors to make sure others understand our methodology and thus our title choice. Can the reviewer explain further their confusion so we know how to do so?
>
> __Graph-tools dependency__: The graph-tools package we use is actively maintained, and more information about it can be found [here](https://git.skewed.de/count0/graph-tool). However, we acknowledge that requiring users to use two different packages for model training vs. graph analysis is burdensome. We will replace the graph-tools package in our codebase with networkx to avoid this conflict. We appreciate the reviewer’s feedback and concern for long-term maintenance.
>
> __Code license__: Our tool is intended to be used for academic research purposes, so we believe the MIT license is appropriate.

---

> > ### Comment · Reviewer_8bW3 · 2022-08-17
> > **Response to the response of authors**
> >
> > Thank you for taking the time to respond to my comments. Below are my comments to your response, following your structure.
> >
> > **Scope of paper is too narrow**
> >
> > I'm positive about the quality and relevance of the work made in [31], and appreciate the efforts put forward to find a way to distribute the datasets, but I don't find this argument relevant to justify the narrow scope of this paper. From your response, it seems that this contribution would be more relevant as supplementary materials for [31] rather than as a publication in this track.
> >
> > To complement this point, one main issue is that this contribution is not a dataset but rather a methodology for the generation of datasets, meaning that all requirements to ensure the quality of the datasets (documentation, quality, maintenance, availability, etc.) cannot be evaluated, as it is on the side of the user that the dataset is generated. Similarly, this contribution is not a benchmark, as no evaluation for example of the success of backdoors attacks is proposed. While I believe this work, with [31], could be leveraged to propose to the community either a standardized dataset or a framework for benchmarking backdoor attacks, my opinion is that the scope of this contribution is currently too narrow to be relevant in this particular track for the ML community.
> >
> > **Threat model not well-explained**
> >
> > Thank you for your proposal to add more context about the threat model. It would be appreciated to be able to review this addition before acceptance, to understand how this will be relevant for researchers working on the security of machine learning models. In particular, a discussion about the suitability of the datasets with respect to real-world situations (do the datasets, despite their limited size, are a good proxy for real situations) would be very valuable.
> >
> > **Title does not match paper**
> >
> > Your response does not address my comment. Here are the three reasons I mentioned in my review, with additional comments:
> >
> > 1. *It is not a dataset but rather a methodology.* The title gives the impression that the contribution of the paper is a set of datasets with natural backdoors, which is not the case.
> > 2. *It is limited to multi-object image datasets* Natural backdoors could be extended to other types of data, such as text, sound, tabular data, etc., even if this extension may not be trivial. However, this work specifically focuses on multi-object image datasets, and the title does not acknowledge this.
> > 3. *The term "natural backdoor" is ambiguous and does not describe well the hypothesis on which the authors rely to construct their methodology* While I understand the term is well-defined in the paper, this is an accepted term in the community that would clear the confusion stemming from its ambiguity, unless I'm mistaken. There are different interpretations of what a natural backdoor could be, for instance it could be a backdoor trigger purposefully put on the image (either physically or on the image itself) that would naturally fit in the context of the image.
> >
> > Therefore, the title does not give in my opinion a good indication of the purpose and scope of the contribution, and should be in my opinion revised. A proposal would be "A graph-based methodology to generate datasets with physical backdoors from multi-object image datasets". I agree that it would make the title longer and less appealing, but I find the current title confusing and misleading.
> >
> > **Graph-tools dependency**
> >
> > Thank your for your response. It is true that the package 'graph-tool' is actually maintained and works well with recent Python version, sorry for the mistake. Regarding the setup, I was able (after a few iterations) to run the graph analysis in a virtual environment. I would recommend to make the generation of the datasets as a standalone tool, decoupled from the training of models, in line with the contribution of the paper.
> > Switching to networkx sounds less critical now but could still be valuable for the user experience, at the condition that it does not affect the performance of the methodology.
> >
> > **Code license**
> >
> > I do not have any concern about the choice of the license, but rather on the fact that some parts of the code seem to come from other projects, with a different license.
> >
> > **Additional comments**
> >
> > As it has been suggested by a reviewer, it would be interesting to get access to a set of datasets (even limited in size) with different natural backdoor triggers to evaluate the types of outcomes of the methodology. Right now, the reader has to set up the code which is currently a bit tedious as it requires compiling external libraries, downloading large datasets, and running the script. This could be done on a ad-hoc toy dataset without copyright issue that does not need to be for a specific task.

---

> > > ### Author Response · Authors · 2022-08-22
> > > **Addressing scope, threat model and title concerns**
> > >
> > > We thank the reviewer for their further detailed thoughts and respond to them below.
> > >
> > > **Scope of paper**
> > >
> > > We appreciate the reviewer’s points on the interpretation of submissions to this track relying on either a self-contained dataset or an analysis of benchmarks. However, we believe our paper is well within the scope of what the “Datasets and Benchmarks” track is intended for as the blog post introducing the track in 2021 (see [here](https://neuripsconf.medium.com/announcing-the-neurips-2021-datasets-and-benchmarks-track-644e27c1e66c)) explicitly states that contributions which analyze existing datasets and provide novel approaches to solving data-related problems in the ML community (as our work does) are welcomed and encouraged. First, the post states that “... we welcome submissions that detail **advanced practices in data collection and curation that are of general interest** even if the data itself cannot be shared. **Audits of existing datasets**, or systematic analysis of existing systems on novel datasets **that yield important new insight are also in scope**.” Furthermore, this introductory post anticipates that publications in this track will “... be a rich body of publications around topics such as new datasets and benchmarks, **novel analysis of datasets and data curation methods**, evaluation and metrics, and societal impacts such as ethics considerations.” Based on this definition of the track’s scope alone, we believe our work is well-suited for this venue.
> > >
> > > We did debate the proper venue for this work, and this blog post detailing the nature of the Datasets and Benchmarks track convinced us to submit to this track instead of the main conference. We also posit that this paper stands on its own compared to [31] due to the shift in perspective from just collecting datasets to *providing an automated tool for dataset generation* as well as the complexity of the techniques used to enable this automation. Our method of using co-occurrences to find viable triggers in existing datasets not only substantially expands the ability of researchers to investigate and mitigate physical backdoors, it may also be of independent interest in non-adversarial settings.
> > >
> > > If the reviewer has any further suggestions for additional writing changes or experiments we can provide that would emphasize the appropriate scope of our paper, we would welcome them.
> > >
> > > **Threat Model and Dataset Size**
> > >
> > > We are working on a revised version of the paper, including an updated description of the threat model, and plan to have that revision available to reviewers by Thursday of this week.
> > >
> > > The sizes of the datasets’ generated from our method for the two source datasets we consider is not a concern, as our datasets are of the same size or bigger than [31] . This is due to the inherent difficulty of collecting large and diverse physical backdoor datasets. Each data point is usually an image that needs to be manually taken, and each additional class requires access to a new physical object or environment. As we discovered in [31], even collecting a small dataset with ~3000 images and 10 classes took several months, due to issues like data quality, pre-processing and IRB approvals.  We will discuss these more nuanced points in the updated threat model.
> > >
> > > **Title**
> > >
> > > We appreciate the reviewer’s points on this issue. We are open to changing the title to “*Finding Naturally-Occurring Backdoors in Image Datasets*.” However, our concern remains that this new title will obscure the main contribution of the paper, which is the ability of our method to generate, given any dataset with multi-labels, a number of viable natural backdoor datasets.
> > >
> > > **Sample datasets**
> > >
> > > We will provide links to 6 standalone, downloadable sample datasets (taken from Table 2 in the paper), along with documented backdoor performance as part of the revision. We appreciate that several reviewers have raised this concern, and while we still emphasize the ability to generate these datasets as our main contribution, we agree that these datasets may aid researchers.
> > >
> > > **Graph libraries**
> > >
> > > As suggested by the reviewer, we have added support for the networkx library along with graph-tool. Initial experiments suggest that the results are similar with both libraries. We also clarify that the graph analysis and model training *are decoupled* in our code base. Although both can be run via the main.py script in our code, they do not need to be run at the same time. A user can easily utilize the graph portion of our codebase to find viable triggers, poisonable classes and their associated statistics without ever training a model.

---

> > > > ### Comment · Reviewer_8bW3 · 2022-08-26
> > > > **Closing the discussion**
> > > >
> > > > Thank you for all your detailed explanations and for addressing my issues swiftly and carefully.
> > > > At this stage, I don't have any additional comments to make. I will read your revised version and update my initial review based on the changes you made.

---

### Official Review · Reviewer_2HBv · 2022-07-26
**Natural Backdoor selected through multi-label**

**Rating:** 6
**Confidence:** 3

**Strengths:**

1. This paper selects the hidden trigger in the open source dataset through multi-label data, which greatly improves the method of obtaining attack data in the previous physical Backdoor method.
2. This paper ensures the co-occurence of trigger and other types of data by designing measures.
3. This paper has done sufficient data experiments to verify the existence and feasibility of Natural Backdoor

**Weaknesses:**

1.Lack of standardized dataset. As this paper is in the track of Dataset & Benchmark, an accessible dataset is needed for fair evaluation.
2.Although the method of establishing the backdoor in this paper is different from the previous physical backdoor, it does not compare with the attack situation after establishing the backdoor before.
3.In sec 5.4, some backdoor defense method are not considered. [1] can identify the backdoor data. [2,3,4] can mitigate backdoor attack. However, the above backdoor defense methods are not used to make experiments to show whether natural backdoor can be identified or mitigated.
4.The performance of trigger selected for different measures is not stable enough in the two data sets in Fig. 5,13. Whether it is necessary to select appropriate selection indicators for different data sets

[1] Guo J, Li A, Liu C. Aeva: Black-box backdoor detection using adversarial extreme value analysis[J]. arXiv preprint arXiv:2110.14880, 2021.
[2]Huang K, Li Y, Wu B, et al. Backdoor defense via decoupling the training process[J]. arXiv preprint arXiv:2202.03423, 2022.
[3]Wu D, Wang Y. Adversarial neuron pruning purifies backdoored deep models[J]. Advances in Neural Information Processing Systems, 2021, 34: 16913-16925.
[4]Li Y, Lyu X, Koren N, et al. Neural attention distillation: Erasing backdoor triggers from deep neural networks[J]. arXiv preprint arXiv:2101.05930, 2021.


**Additional Feedback:**

No.

**Clarity:**

The citation in the supplementary material is false. For example, wrong citation in line 483 "[18] uses light-based reflections as backdoor triggers."

**Correctness:**

Yes, this paper selects the data that meet the selection criteria from the multi-label open source data set and then modifies the labels of these data.

**Documentation:**

Yes, the documentation is sound.

**Ethics:**

No.

**Relation To Prior Work:**

Yes, this paper compares the previous setting method of physical backdoor and some other backdoor method. However, the article lacks the introduction of the backdoor defense and some related work. The backdoor defense method in Sec 5.4 is lacked.




**Summary And Contributions:**

The paper proposed a method to obtain the Natural Backdoor Dataset by modifying the labels of specific samples in the open source dataset. The paper develops some criteria to select appropriate multi-object pictures to ensure that the trigger can be recognized and triggered by multiple classes. The paper proves the existence of trigger selected according to the measures and provides the results of the post-attack model and backdoor defense against natural backdoor.

---

> ### Author Response · Authors · 2022-08-11
> **Response to Reviewer 2HBv**
>
> We thank the reviewer for their comments, and we respond to their concerns below.
>
> __Lack of standardized dataset__: We are concerned that if we provide a single dataset, future work will focus on evaluating only against this dataset rather than engaging more fully with our techniques. Instead, in our supplementary materials, we provide a comprehensive code base enabling both recreation of our key results (e.g. Figures 5, 13) and future investigation and extensions of our method. By running our provided scripts, any user can recreate both our results and the datasets used in our evaluation. Can the reviewer clarify what they had in mind re: a standardized dataset, if our current code base does not satisfy this? We want to be sure we are providing helpful materials to readers.
>
> __“Although the method of establishing the backdoor in this paper is different from the previous physical backdoor, it does not compare with the attack situation after establishing the backdoor before.”__ We are confused by this statement. Can the reviewer please clarify what they mean? Are you suggesting we compare our current results with results from [31], the prior physical backdoor attack paper? If so, we are happy to add a comparison table to [31]’s results on object recognition in a camera-ready version of the paper, if it is accepted.
>
> __Critique of defense evaluation__: The goal of this paper is to evaluate whether our proposed natural backdoor generation technique produces physical backdoors equivalent to those introduced in [31]. In making this comparison, we evaluated 4 well-known, highly-cited defenses known to fail for physical backdoors. It is not the goal of this paper to evaluate whether natural backdoors are vulnerable to any backdoor defense, but primarily rather that they are at least as robust as physical backdoor attacks.
>
> That being said, we are happy to evaluate additional defenses – [NNoculation](https://dl.acm.org/doi/abs/10.1145/3474369.3486874) (AiSec ‘21) and [SentiNet](https://arxiv.org/pdf/1812.00292.pdf) (DLS ‘20) – in the camera-ready version of the paper, if accepted. We selected these defenses because they do not make strong assumptions about the nature of the backdoor trigger, which may make them more effective against natural backdoors. Evaluation of these is ongoing, and we will post results as a comment when we have them.
>
> __Performance of trigger is not stable (cf. Figures 5, 13)__: We acknowledge that different triggers yield different backdoor performances, although attack success rate always remains relatively high. However, this variation drives home the point of our proposed methodology – to provide users with a wide set of physical trigger options and empower them to explore which factors make them more or less successful.
>
> __Citation in supplementary is wrong:__ We apologize for this error, and we will correct it in the camera-ready version of our paper, if accepted.
>
> __Lacks introduction of backdoor defense:__ We can add additional descriptions of proposed backdoor defenses in Section 5.4, before our evaluation.

---

> > ### Author Response · Authors · 2022-08-29
> > **Final questions from Reviewer 2HBv?**
> >
> > Dear Reviewer 2HBv,
> >
> > We hope you had a chance to look at our responses to your questions and the revised version of the paper we have submitted. Please let us know if you have any additional questions or concerns that we can address at this stage.

---

> > > ### Comment · Reviewer_2HBv · 2022-08-29
> > > **Response to the revised paper about backdoor defense**
> > >
> > > Thank you very much for solving my problems in revised paper. You answered a lot of my questions in great detail. I have just one question for the backdoor defense outcome:
> > >
> > > I mentioned a couple of defenses earlier that were equally good, why didn't you compare them in this rebuttal period? As you mentioned in line 775 of the supplementary material, the hypothesis that NNocluation needs a clean dataset has been mentioned in many works on backdoor defense. I think it is also relatively reasonable to large data sets.
> > >
> > > Thank you very much for your response.

---

> > > > ### Author Response · Authors · 2022-08-29
> > > > **Re: backdoor defense**
> > > >
> > > >
> > > > Thank you for your response. We emphasize that the goal of our paper is __to develop a new approach enabling researchers to build low-cost physical backdoor datasets__. Therefore, we evaluate attacks/defenses as examples to demonstrate the utility of our approach and to validate that datasets obtained via our method still provide value for researchers experimenting with physical backdoor attacks.  Thus, the goal of our paper is __not__ to exhaustively compare against against existing backdoor defenses.
> > > >
> > > > However, per yours and other reviewers’ requests, we did evaluate a few additional defenses, bringing the total of defenses evaluated up to 6 — a large number. We selected and evaluated the additional defenses we did because they made few assumptions about the nature of the backdoor trigger in question or its impact on the model, since prior work  showed that physical backdoors violate traditional backdoor defense assumptions. Overall, we believe that our defense evaluation goes above and beyond the initial goal of comparing natural backdoors to physical backdoors, and leaves open the possibility of much future work on the subject of natural backdoors (including tests of different defenses).
> > > >
> > > > Regarding the specific defenses you mentioned — we apologize for misinterpreting your initial review.  We read your initial comments and citations as a broader suggestion that we should evaluate more defenses, not specific instructions that we evaluate those defenses. Consequently, we did not evaluate these in addition to the other defenses we identified (SentiNet and NNoculation). However, we can add evaluation of these defenses to the camera-ready version of the paper, if the reviewer believes they will add value to the paper.

---

### Official Review · Reviewer_cMaN · 2022-07-28
**Valuable work but need improvements**

**Rating:** 4
**Confidence:** 4
**Correctness:** Correct.
**Clarity:** could be improved.

**Strengths:**

This idea alleviates the high trigger generation costs in physical backdoors is good.



**Weaknesses:**

This paper lacks a comparison with past physical attack methods.

The number of defense methods used in this paper is small, and these methods are old.

This paper does not include experimental results for the other two scenarios declared in the section "Other usage scenarios", which makes us unsure whether the algorithms combine well with other physical attacks in more general cases.


-----

Considering there is one month for rebuttal, I don't think the authors have made good efforts to make this work better. Ignoring the reviewers suggestions is not a smart rebuttal strategy.

I also carefully read other reviewers' comments and the authors' responses. I get a similar opinion, and several important and good suggestions/concerns are not followed/addressed.

I keep the initial score.

**Additional Feedback:**

no

**Documentation:**

yes

**Ethics:**

no concerns

**Relation To Prior Work:**

yes

**Summary And Contributions:**

This paper presents a novel idea for facilitating the generation of physical backdoors. Using many co-occurring objects in a public dataset as triggers significantly reduces the cost of generating physical backdoors. Using graphs to represent the occurrence relationships between objects and find important nodes and corresponding MIS shows promising results.

---

### Official Review · Reviewer_17uL · 2022-07-31
**Method to automatically generate physical backdoor attack datasets**

**Rating:** 8
**Confidence:** 4

**Strengths:**

* This is an important and relatively unexplored area of work (physical triggers for backdoor attacks).
* The work has a clear intuition.
* The design using network centrality measures is well done.
* The evaluation seems to back up the intuition.


**Weaknesses:**

* I would have liked to see further discussion of how applicable this technique is to the more general class of backdoor attacks and defenses, beyond physical triggers.
* It was not clear to me how the work generalizes to more number of trigger classes of objects.


**Additional Feedback:**

What about backdoor defenses against non-physical triggers? Do they transfer over well to the physical world backdoors? [30] is a representative sample of this. What is fundamental about these triggers that escape detection by defenses?

Lines 167-172: You could consider multiple trigger objects rather than a single one. What is the interaction of these multiple triggers? Is it always best to pick the top-k or combinatorial effects of two (or more) trigger classes that are individually lower ranked is more significant?

What is the relative contribution of these two factors in choosing the trigger class --- centrality and MIS? While an experiment is given (Table 1, lines 270-275), some intuition about what helps how much would be good.

There is good comparison of performance of existing backdoor defenses against their generated datasets.


**Clarity:**

The paper is well structured and well written.


**Correctness:**

The dataset is soundly constructed. There is good support for reproducibility through the run_on_gpus_centrality_ablate.py script that is set up to reproduce the results from Figures 5 and 13 in the paper. I wondered about reproducibility for Figures 6 and 7 though.

For ease of use, the authors should provide a repository with images with the trigger objects inserted, so that one can readily train a model using the dataset.


**Documentation:**

The Github repo is well done. See above (under "Correctness") for further usability of this resource.


**Ethics:**

No ethical concern.

**Relation To Prior Work:**

This is a relatively new area. The most closely related work is from the authors' group: [31].

There are a few other works in backdoors in images that the authors can discuss [A][B].
[A] Li, Yiming, Tongqing Zhai, Baoyuan Wu, Yong Jiang, Zhifeng Li, and Shutao Xia. "Rethinking the trigger of backdoor attack." arXiv preprint arXiv:2004.04692 (2020).
[B] Lin, Junyu, Lei Xu, Yingqi Liu, and Xiangyu Zhang. "Composite backdoor attack for deep neural network by mixing existing benign features." In Proceedings of the 2020 ACM SIGSAC Conference on Computer and Communications Security, pp. 113-131. 2020.


**Summary And Contributions:**

The paper deals with physical triggers for backdoor attacks. Physical triggers are real-world objects present in images at their creation and such backdoors have been shown to successfully evade existing defenses for object and facial recognition [31-CVPR21] (work from the same group). This paper deals with a cheap way to generate datasets with physical triggers. The key insight is to use existing widely used image datasets and relabel some trigger objects in them.

---

> ### Author Response · Authors · 2022-08-11
> **Response to Reviewer 17uL**
>
> We thank the reviewer for their helpful feedback and respond to their points below.
>
> __Backdoor defenses against non-physical triggers transfer?__ We do test against Neural Cleanse [30], see Table 3. As [31] explored, defenses against non-physical triggers tend to fail against physical triggers because they make assumptions about trigger construction/behavior that do not hold up in the physical trigger setting. For example, [30] assumes that triggers represent a minimal perturbation from one class to another. However, physical triggers can be arbitrarily large (e.g. a pair of jeans), so this assumption does not hold up and [30] fails to mitigate natural backdoors.
>
> __Multiple trigger objects:__ Considering multiple trigger classes would be an interesting extension to this work. It seems the crux of the issue here would be finding triggers with large-enough overlapping maximum independent subsets. We are currently evaluating the possibility of using multiple trigger classes, and we will share our preliminary results in a follow-up comment. If the results are reasonable, we will add them to the camera-ready version of the paper, if accepted.
>
> __Relative contribution of centrality vs. MIS:__ We thank the reviewer for this question. We show in Table 1 that MIS is more important than centrality in identifying effective triggers. However, we can enhance our discussion of the relative importance of centrality vs. MIS to a revised version of the paper to include the following points. In our experiments, we observe that different centrality measures capture different structures in the datasets, highlighting different potential trigger/class sets. However, it is the MIS step that ensures dataset quality, regardless of the centrality method used to find it. By using MIS, we minimize overlap between the clean classes connected to the trigger, ensuring higher model accuracy.
>
> __Discuss additional backdoor attacks:__ We can add references to different backdoor attacks you mention in our background section. We also plan to evaluate against two additional defenses – NNoculation (AiSec ‘21) and SentiNet (DLS ‘20). We can share these results in a comment and add them to a camera-ready version of the paper, if accepted.
>
> __Reproducibility:__ We are happy to add a script to reproduce Figures 6 and 7 to the repository. Regarding the comment about providing a repository with images: this is exactly what we do. After cloning our repository, users can create their own natural backdoor datasets from existing datasets, with ImageNet and Open Images being supported at the moment. Using our codebase, this dataset creation process is simple, and we include a README to ensure users understand what to do.
>
> It seems the reviewer is suggesting that we select a particular trigger/centrality metric and provide the physical backdoor dataset for this parameter setting. We could do this if the reviewer would like us to, but we do not think this would be particularly helpful to the research community. First, providing a single dataset would put unnecessary focus on a particular natural backdoors parameter setting, when in reality there are multiple settings that work well. Second, we are concerned that if we provide a single dataset, future work will focus on evaluating only against this dataset rather than engaging more fully with our techniques.

---

> > ### Author Response · Authors · 2022-08-29
> > **Final questions from Reviewer 17uL?**
> >
> > Dear Reviewer 17uL,
> >
> > We hope you had a chance to look at our responses to your questions and the revised version of the paper we have submitted. Please let us know if you have any additional questions or concerns that we can address at this stage.

---

### Author Response · Authors · 2022-08-24
**Summary of Revised Paper**

We thank the reviewers for their thoughtful and constructive engagement with the paper. As reviewers ourselves, we greatly appreciate the reviewers’ efforts at providing thorough and insightful commentary on the paper.

We are glad to hear that reviewers think our methods *address an important area of research* ([17uL](https://openreview.net/forum?id=v3yM5zVzP4C&noteId=e0717oK3uQ)), *have clear intuition* ([17uL](https://openreview.net/forum?id=v3yM5zVzP4C&noteId=e0717oK3uQ)), and *provide a useful, versatile tool* for backdoor attack researchers ([8bW3](https://openreview.net/forum?id=v3yM5zVzP4C&noteId=seH1SrCozrW), [jCiv](https://openreview.net/forum?id=v3yM5zVzP4C&noteId=GuTze4fQHv1)), while *greatly improving upon the previous method* ([2HBv](https://openreview.net/forum?id=v3yM5zVzP4C&noteId=rhb1TBxOLdp)). Given our extensive experiments, we appreciate that the reviewers found our work to be *well-designed* ([17uL](https://openreview.net/forum?id=v3yM5zVzP4C&noteId=e0717oK3uQ)), with *carefully-assessed soundness* ([8bW3](https://openreview.net/forum?id=v3yM5zVzP4C&noteId=seH1SrCozrW)) and *good evaluations* ([jCiv](https://openreview.net/forum?id=v3yM5zVzP4C&noteId=GuTze4fQHv1)). Finally, given the importance of providing open-source code to ensure reproducible research, we are pleased to hear that reviewers believe our GitHub repo is *well-documented* ([17uL](https://openreview.net/forum?id=v3yM5zVzP4C&noteId=e0717oK3uQ), [8bW3](https://openreview.net/forum?id=v3yM5zVzP4C&noteId=seH1SrCozrW)).

Based on our dialogue with the reviewers over the last few weeks, we have revised and updated our paper in accordance with their comments and suggestions. The revised paper and supplementary materials have been uploaded, with updated text in green. Should the paper be accepted, we are committed to further polishing this revised version and making any additional changes requested by the reviewers. The changes made in the revision, along with where the changes can be found, are summarized below. We highlight in __bold__ changes that address issues raised by multiple reviewers.
- __Evaluation of additional backdoor defenses__ (Appendix G).
- Updated paper title to “Finding Naturally-Occurring Physical Backdoors in Image Datasets”.
- Clarified threat model (Section 2).
- __Added description of backdoor defenses__ (Section 2).
- Added more straightforward examples to “other usage scenarios”  (Section 4).
- __Added a comparison to the object recognition model performance of [original physical backdoors paper](https://arxiv.org/abs/2006.14580)__  (Section 5.3).
- Added a description of the multi-trigger scenario and statistics about possible multi-triggers in Imagenet and OpenImages
- Updated captions of all tables to be above instead of below.

Additionally, based on feedback from reviewers, we made the following changes to [our codebase](https://github.com/uchicago-sandlab/naturalbackdoors):
- __Added example datasets from Table 2 in our paper in a linked Google Drive folder__ (see README).
- __Added reproducibility scripts for Figures 5, 6, 12 and 13, as well as Tables 3 and 9.__
- Replaced the graph-tools package with networkx.
- Switched to using a single conda environment rather than two.

---

### Meta-Review · Area_Chair_PF8K · 2022-09-09

**Recommendation:** Accept
**Confidence:** 3

**Metareview:**

The main contribution of the work is a versatile tool to generate natural backdoor datasets that can be used to benchmark future models and backdoor defenses. The reviewers acknowledge that having a low-cost approach to generating these physical backdoor datasets is important since there is a typically a high cost to introduce physical triggers in realistic training data.

However, some concerns were raised over the fact that the authors provide a technique to generate natural backdoor datasets, but do not curate a standardized benchmark for the community.  Additionally, the authors do not use human verification of labels, which some reviewers argue harms the credibility of the constructed datasets.

The authors do release sample datasets for the community, which I think addresses the concern regarding lack of benchmark datasets. I would additionally encourage the authors to spend effort validating that the techniques to generate such datasets are producing sensible labels, so that the community has increased trust in both the tool and the released benchmarks.

---

### Decision · Program_Chairs · 2022-09-16

Accept